# Applications and Current Medico-Legal Challenges of Telemedicine in Ophthalmology

**DOI:** 10.3390/ijerph19095614

**Published:** 2022-05-05

**Authors:** Daniela Mazzuca, Massimiliano Borselli, Santo Gratteri, Giovanna Zampogna, Alessandro Feola, Marcello Della Corte, Francesca Guarna, Vincenzo Scorcia, Giuseppe Giannaccare

**Affiliations:** 1Department of Surgical and Medical Sciences, University ‘Magna Græcia’ of Catanzaro, Viale Europa, 88100 Catanzaro, Italy; danielamazzuca7@gmail.com (D.M.); gratteri@unicz.it (S.G.); dellacortemarcello@gmail.com (M.D.C.); francescaguarna@blu.it (F.G.); 2Department of Ophthalmology, University ‘Magna Græcia’ of Catanzaro, Viale Europa, 88100 Catanzaro, Italy; mborselli93@gmail.com (M.B.); vscorcia@unicz.it (V.S.); 3Department of Law, Economics and Human Sciences (DIGIES), Mediterranea University of Reggio Calabria, Via dell’Università 25, 89124 Reggio Calabria, Italy; giovanna.zampogna@unirc.it; 4Department of Experimental Medicine, University of Campania ‘Luigi Vanvitelli’, Via Luciano Armanni 5, 80138 Naples, Italy; alessandro.feola@unicampania.it

**Keywords:** ophthalmology, telemedicine, eye, medico-legal, COVID-19, communication, privacy, clinical-practice

## Abstract

Background: The digital revolution is redesigning the healthcare model, and telemedicine offers a good example of the best cost-effectiveness ratio. The COVID-19 pandemic has catalysed the use of the telemedicine. The aim of this review is to describe and discuss the role and the main applications of telemedicine in the ophthalmic clinical practice as well as the related medico-legal aspects. Methods: 45 original articles and 5 reviews focused on this topic and published in English language from 1997 and 2021 were searched on the online databases of Pubmed, Scopus, Web of Sciences and Embase, by using the following key words: “telemedicine”, “privacy”, “ophthalmology”, “COVID-19” and “informed consent”. Results: Telemedicine is able to guarantee patient care using information and communication technologies. Technology creates an opportunity to link doctors with the aim of assessing clinical cases and maintaining high standards of care while performing and saving time as well. Ophthalmology is one of the fields in which telemedicine is most commonly used for patient management. Conclusions: Telemedicine offers benefits to patients in terms of saving time and costs and avoiding physical contact; however, it is necessary to point out significant limitations such as the absence of physical examinations, the possibility of transmission failure and potential violations of privacy and confidentiality.

## 1. Introduction

The World Health Organization (WHO) defines telemedicine as: “The provision of health services, where distance is a critical factor, by all health professionals who use information and communication technologies for the exchange of valid information for the diagnosis, treatment and prevention of diseases and injuries, research and evaluation, and for the continuous training of health workers, all in the interest of advancing the health of individuals and their communities” [1]. Nowadays, telemedicine is widely used and has a great utility in the management of patients in areas with poor health services, in conflict-affected countries or in refugee populations [2]. Since there are multiple types of network access, including telephone lines, local computer networks, television cable, optical fibers and low-altitude orbiting satellites, no area of the world is too remote or too poor to use this service. Recent studies have shown that telemedicine, if used by the community, can lead to a significant improvement in patient management, long-term cost savings and greater sustainability for public health, even in developed countries. The current modalities for performing teleconsultation are: (i) real time, in which doctor and patient can interact with each other at the time of consultation and (ii) synchronous, where the patient answers question and the doctor responds later, according to his convenience via short message service (SMS) or e-mail [3,4,5,6,7]. The current COVID-19 outbreak has further reinforced the need for the use of remote consultation in order to reduce contact between patients and healthcare professionals to the bare minimum [3]. In fact, the fear of being infected by COVID-19 has drastically reduced access to hospitals in favour of telematic consultations between patient and doctor. This has favored the reduction of infections in most hospital centers, contributing to the expansion of health networks for consultation via telematics. Nowadays, in the United States about 20% of the beds of the adult intensive care unit are supported by some form of tele-intensive care coverage [4]. Tele-technology applied to medicine is rapidly creating new fields such as tele-oncology, tele-cardiology, tele-dermatology and tele-radiology [1,8].

The goal of this review is to highlight the potential role of telemedicine as a future resource in the diagnosis and management of various ocular disorders, and to further compare the legal regulations in force in different countries.

## 2. Methods

The present research was carried out on scientific papers published between 1997 and 2021. Pubmed, Scopus, Embase and Web of Science were searched for integrative reviews and original papers. Thirty-seven original articles and four reviews were chosen from Pubmed, two scientific articles and one review from Scopus, and two and four papers from Embase and Web of Science, respectively. The search in the database was carried out using the following keywords: ‘‘telemedicine’’, ‘‘privacy’’, “ophthalmology”, “COVID-19” and “informed consent’’. The inclusion criteria represented articles from the medical literature dated in the selected time range, published in international journals, cited at least five times in other articles and relevant to the chosen topic under consideration. Outcomes of interest included telemedicine performance during the COVID-19 pandemic period, patient compliance and satisfaction, economic impact on the healthcare system and legal implications. A total of 73 integrative review publications for potential inclusion were identified. The authors operated the preliminary evaluation independently by reading the abstracts and drafting a list of the articles considered eligible. Then, the lists were compared, and in case of discrepancies, a consensus was reached. We reviewed all records manually to verify and remove duplicates that had not been previously detected. A reviewer screened all titles and abstracts of the retrieved citations. We obtained and independently examined the full-text copies of potentially eligible reports for further assessment. We have evaluated common parameters for each article in order to make the selection as appropriate as possible: relevance of the study, originality of the study, strengths and weaknesses of the study, clarity and constructiveness. For each parameter, we attributed a value from 1 to 5, for a maximum value of 25. Only studies with a score > 15 were chosen [9]. The evaluation was conducted independently and opinions compared to create a consent [1]. By using these criteria, 45 original articles and 5 reviews were selected and a careful narrative overview of these papers was conducted.

## 3. Results

### 3.1. Telemedicine in Different Medical Specialties

Telemedicine allows the transfer of medical information (texts, sounds, images, etc.) for preventing, diagnosing, treating and monitoring patients in a wide range of medical and surgical specializations. In surgical specialties, one of the main innovations concerns remote-guided surgery that allows for remote operation or to train the procedural skills of surgeons through mentors from distant places. This is favored by the increase in speed and the improvement of the quality of data transmission and connection. In a systematic review of 24 studies using telecommunications technologies in the field of surgical care, Asiri and collaborators concluded that the use of telemedicine had beneficial effects in pre- and post-operative examinations, including follow-up visits [2]. It was also particularly appropriate to use teleconsultation to provide patients with post-operative care for low-risk elective procedures [3,10].

In recent decades, teleradiology has greatly expanded to cover a wide range of radiological practices. According to the American College of Radiology: “*teleradiology is an electronic transmission of radiological images from one place to another for interpretation or consultation purposes*” [10]. Teleradiology can help increase imaging efficiency and eliminate geographical and temporal discrepancies in the evaluation of patient examinations. Technological limitations and regulatory brakes hinder the optimal practice of teleradiology [11,12].

General practitioners regularly need specialized advice for therapeutic adaptation, interpretation of the electrocardiogram or to facilitate referral to the local specialist [13,14,15,16]. Ultrasound is an effective diagnostic tool and its use within telemedicine (“tele-ultrasound”) has been increased considerably in recent years, particularly in developed contexts; conversely, the usefulness of tele-ultrasound in developing countries is less well established.

Regarding tele-cardiology, smartwatches are becoming increasingly popular and are increasingly used to monitor human health. Inherited and acquired long QT interval is a risk marker for potential severe cardiac arrhythmias and sudden cardiac death. The possibility of evaluating the QT interval using a commercially available Apple Watch has been studied. The reliability of QTc measurements with smartwatches has been demonstrated by confirming that a smartwatch can easily and reliably evaluate the QT interval [17]. Another important use of tele-cardiology concerns the emergency. It is now routine to perform an electrocardiographic trace during the journey by ambulance, in patients who have had an acute heart attack. There are systems composed of a Hub-and-Spoke network of hospitals and ambulances that ensure a quick exchange of information allowing STEMI patients to be managed in the shortest time [13,14,15]. It is useful to have networks based on pre-hospital triage of acute myocardial infarction using telemedicine electrocardiograms to manage the patient in a more controlled way [16]. Telemedicine appears to be a cost-effective modality for the management of stroke [18].

Dermatology is a field with few life-threatening conditions. Short delays in the diagnosis and treatment of skin conditions rarely impair the patient’s long-term health. Several skin conditions are present for a long time and patients need repeated follow-up visits. Due to the highly visual nature, the majority of skin conditions can be detected by means of a picture, especially if an accurate history is collected. In 2004, tele-dermatology (“Tele-Derm”) was established as an online consulting service, combined with a central portal for online dermatological education, resources, links, discussions and professional development activities. Often the dermatologist receives private photographs from the patient in order to diagnose or frame a clinical picture and set up therapy [19,20].

### 3.2. Focus on Telemedicine in Ophthalmology

Ophthalmology is a field in which telemedicine has been used for some time, having quickly adapted to the digital world in the use of advanced diagnostic and treatment technology systems, and being a specialty based on visual evaluation. A fundamental part of the ophthalmologist’s work is instrumental diagnostics, which is increasingly digitized nowadays. It is also routine and possible in different centers to acquire images to evaluate the patient’s follow-up over time.

Ophthalmological applications (apps) are transforming smartphones into medical devices; currently, there are 121 apps dedicated to ophthalmology that can test vision using the Snellen visual acuity test, including in the case of children and illiterate people. Some of the apps can evaluate color vision, astigmatism, pupil size, the Amsler grid test, oculomotor reflex, the Worth 4 dot test and accommodation target, the red desaturation test and an optokinetic drum simulator. Smartphones usig a pinhole adaptor can be used to quantify the refractive error without significant difference compared to subjective refraction collected during a routine visit. There are several photo-adapters available for smartphones which make them useful ophthalmic devices for taking images of both anterior and posterior eye segments. There are adapters designed to attach to the “PanOptic Ophthalmoscope” for acquiring fundus photos through undilated pupil. It is also possible to take high quality pictures of retinas using smartphones and indirect lenses. The examiner can watch the display and evaluate the real-time images of the eye with other practitioners, and record and send the findings. 

Telemedicine can be applied for preventive purposes by providing patients with information on their health conditions and by monitoring chronic diseases such as diabetic retinopathy (DR), glaucoma, age-related macular degeneration (AMD), etc. Among the main problems with the use of tele-ophthalmology is the unfamiliarity with its use for both the patient and the doctor. Physician training and easy-to-use platforms for patients can overcome these limitations. Telemedical examination of emergent ophthalmic complaints consisting of a patient questionnaire, anterior segment and fundus pictures and ophthalmic vital signs, may be useful to reliably triage eye disease based on presenting complaints [21]. The main studies dealing with the applications of telemedicine in various ocular disorders are reported in Table 1 [21,22,23,24,25,26]. Due to the frequent presence of opaque dioptric media (e.g., cataract), tele-ophthalmology programs with non-mydriatic fundus camera have a higher rate of non-updatable images. Another fundamental point are the risks related to privacy that include mainly the lack of control over the collection, use and sharing of data. In addition, smartphone apps can capture and disperse sensitive data such as location sensor data without the user’s knowledge. Equipment software and hardware must be updated regularly. Inadequate information and communication technology infrastructure, including poor internet connection, electronic health equipment and insufficient access to electricity for the proper implementation of tele-health services, were recognized as the most common logistic barriers. The lack of availability of ophthalmologist for emergency services causes delays in the start of a therapy and can compromise a long-term prognosis. According to some surveys proposed to patients, it emerged that the most common questions related to ocular signs and symptoms, such as redness, pain, tearing, reduction of visual acuity, and the prescription and how to use the drugs [3,5,22]. The most common advice given to the patient was related to the use of medications. In order for a clinical practice to handle more and more fields in the future, it is necessary to adequately train the specialists in the management of remote cases. The teleconsultation should not be anonymous; both the doctor and the patient must know the identity of each other, but adequate privacy of patients must be ensured [3,5,27,28,29] (Figure 1).

#### 3.2.1. Diseases of the Appendages and Orbit

The diagnosis and management of diseases of the outbuildings and orbit is easily evaluated by videoconference, as it is possible to carry out inspections simply. The ophthalmologist is able to evaluate the outer portion of the eye, eyelids and eyebrows and easily identifies any neoformation, such as cysts, blepharitis and chalazion. By integrating an accused ophthalmological history, it is possible to set up a specific therapy for the present condition and evaluate the evolution of the clinical picture over time, also through the acquisition of serial images [6,20].

#### 3.2.2. Ocular Surface Disease

Ocular surface diseases represent the most common complaints in the ophthalmic practice. The novel habits related to the measures for controlling the pandemic (intense smart working/schooling, routine use of face mask) has further increased their prevalence, and in particular that of dry eye disease [30,31]. Due to the visual signals available during the video consultation, it is possible to diagnose lid abnormalities commonly present in the setting of ocular surface diseases, such as chalazion, hordeolum and blepharitis. However, due to the inability to stain they eye with fluorescein dye, some conditions may be difficult to deal with unless they are highly visible upon external examination. A study conducted for eye surface disorders found that the sensitivity and specificity for the diagnosis of conjunctival hemorrhage, local degeneration of the cornea and eyelid tumors are equal to 100% [6,7].

#### 3.2.3. Refractive Error Screening

A study by the University of Arizona used telemedicine to screen the presence of myopia and other refractive errors [27,29]. Among the plenty of health problems resulting from the home confinement of children during the pandemic period, there is the increased risk of myopia (namely, “quarantine myopia”) [30]. Furthermore, near work activities (e.g., reading and writing) are also linked to myopia due to several factors including brightness and chromatic spectrum of light, energy at high spatial frequencies, peripheral defocus and circadian rhythms. Through telemedicine, it has become easier for parents to relate to the ophthalmologist online consultations in order to improve access to healthcare. A first evaluation based on an accurate direct/indirect medical history and the sharing of photographic files helps to prevent and promptly treat any visual defects. Also, digital images of the light reflections of children’s pupils can be collected and transmitted to an ophthalmologist to assess astigmatism and strabismus screening before the conduction of a complete accurate visit [6].

There are several methods to self-manage vision at home—monocular and binocular—but at the moment the most used are those based on specific apps. Smartphone-based visual acuity (VA) apps that can be used in a teleophthalmology portal have been validated. Smartphone-based apps were easy to download and can be used to control patient distance and near visual acuity. An effect of age and refractive error should be considered when interpreting the results [32].

#### 3.2.4. Diabetic Retinopathy

Traditionally telemedicine has been used primarily in ophthalmology for screening and diagnosis of DR. One of the barriers to screening DR via tele-ophthalmology is the high cost of the investments required for the installation of the fundus camera. The successful integration of tele-ophthalmology in DR was hampered due to non-updatable images, poor resolution, small size and opacity. Traditionally, retinal assessment for DR is made using eye drops for dilating the pupil. Following remote eye examination, patients with DR can be referred to an ophthalmologist if further examinations or surgery are necessary. Various studies have reported that telemedicine is a reliable method for DR screening. Digital images are taken through mydriatic retinal cameras, such as stereoscopic images, single still images and color and monochrome photography. In addition, the convenience of non-mydriatic photography for remote operators and patients makes them compelling tools for teleophthalmology. However, digital photographs may be limited in detecting macular edema, neovascularization and other signs of DR [6,7,22] (Table 2).

#### 3.2.5. Retinopathy of Prematurity

Retinopathy of prematurity (ROP) is the leading preventable cause of childhood blindness. In case of prematurity, retinal vascularization is incomplete at birth, leaving avascular areas to form the ROP bed. There are two screening modes: indirect helmet ophthalmoscopy (with 28 or 30 diopter lens) and wide-field digital camera (or wide-field retinophotography). These methods have become the reference (gold standard), because they allow for objective documentation and facilitate diagnosis and follow-up. Recording digital retinophotographies also facilitates ROP screening via telemedicine, especially in hospitals without access to a pediatric ophthalmologist, and enables automated analyses that could become an aid to screening [23,34].

#### 3.2.6. Management of Intravitreal Injections

Intravitreal injections (IVTs) are a fundamental treatment for macular oedema. It is necessary to follow time-dependent therapeutic regimens in order to optimize patients’ visual outcomes. Following the restrictions caused by the pandemic, they have been adapted to a different process of disbursement of IVTs; here, telemedicine has found ample room for adaptation. According to an Italian study, all patients were screened for signs of infection and contacts with suspected or confirmed cases of COVID-19. A first evaluation was carried out by telephone, collecting visual symptoms and complaints of changed vision. Patients were then treated according to three stages of severity of the clinical picture, through a prior clinical evaluation through telephone consultation and study of follow-up through the images acquired in order to limit the risks but at the same time maintain high standards of care [31]. In recent years, the ability to perform OCT at home with a specific device for self-monitoring has been introduced, in the specification for AMD. The new Novel Sparse OCT is a valid and easily available retina scanner. It could be applied as a portable self-measuring OCT system. Its use can facilitate sustainable monitoring of chronic retinal diseases by providing easily accessible and continuous retinal monitoring [24].

#### 3.2.7. Glaucoma

Meta-analysis studies reveal that remote investigations for glaucoma can accurately discriminate the results of tests with a higher probability of positive cases. In a study from the University of Texas, glaucoma screening was conducted using the stereoscopic images of the digital disc taken through a non-mydriatic camera [29]. The ocular tone of the patients examined was evaluated through Tonopen (Tinsley Medical Instruments, Ltd., Croydon, UK). The perimeter fields were acquired at the clinic for review by campus ophthalmologists. Several studies indicate that tele-ophthalmology for glaucoma is almost as effective as in-person visits and improves patients’ participation in screening thanks to the easier examination execution [7,11,35]. In addition to Home-OCT (see Section 3.2.6), there is also the possibility of performing visual field tests for glaucoma screening using a PC monitor or virtual reality glasses at home. The advantages of this system are that it uses the webcam as a “photometer” and validates the reliability of the results at the end of the test. This system also allows the patient to send the results to the ophthalmologist via email and allows the doctor to combine the results of two or more tests into a single test to achieve greater statistical accuracy. The operational characteristics of the receiver (ROC) of this low-cost test show that it is reliable, at least when compared with the Humphrey perimeter, and does not require specialized equipment. The test may be useful for home screening of glaucoma [25].

#### 3.2.8. Management of Keratoplasty

Different indications for keratoplasty exist according to patient’s underlying pathology. Before performing surgery, it is necessary to have the cornea suitable for the type of transplant to be performed. A study highlights the encounter between this type of surgery and telemedicine through determining the feasibility of using telemedicine consultations in the evaluation of recovered donor corneas for transplantability [26]. It suggests that telemedical review of corneal tissue using high-quality digital images may be adequate for the accurate identification of specific corneal outcomes commonly encountered by eye banks. Following surgery, invariably performed in an operative regimen, it is necessary to perform a first follow-up in the present the day after. The following check-up visits may also be performed through telemedicine channels (see Section 3.2) in order to evaluate the conditions of the transplant and the adequacy of topical therapy. In case of suspicion of corneal worsening/rejection/decompensation, an assessment in the presence of the Ophthalmologist becomes necessary [18,21,26].

#### 3.2.9. Emergency

Tele-ophthalmology has a fundamental role in the management of emergency conditions for reaching a quick diagnosis and treatment, especially in rural areas [36]. It often happens that the patient contacts his trusted ophthalmologist by phone when an emergency condition such as instant loss of vision from one eye, unbearable acute pain or an ocular perforating trauma arises. In these cases, the speed of management of the clinical case is fundamental. It is necessary to go to the nearest emergency room in order to be treated according to the guidelines and to manage the case with appropriate medical and surgical therapy [7,37].

#### 3.2.10. Post-Surgical Care

A recent study has shown that there was a reduction of ophthalmology surgery during the pandemic in the lockdown period compared to the intra-year and inter-year control periods: 96.2% and 96.4% for elective procedures, 49.7% and 50.2% for urgent procedures, 48.5% and 48.6% for intravitreal injections, respectively [38]. Almost all of patients undergoing cataract surgery are geriatric and have multiple comorbidities. For post-operative control purposes, video consultation can detect corneal edema and opacifications after cataract surgery; however, there is less evidence on the detection of milder alterations, such as folds of the Descemet membrane [7].

#### 3.2.11. Ophthalmological Consultations

With the outbreak of the COVID-19 infection, the way of doing ophthalmological consultations within the hospital environment has also changed. The greatest interactions for the ophthalmologist in the field of medical consultations take place with neurologists, otolaryngologists, dermatologists, cardiologists, endocrinologists, maxillofacial surgeons and anesthesiologists. In general, it is recommended that the patient is transported from the department where he/she is hospitalized to the eye clinic in order to investigate the diagnostic suspicion with the appropriate tools. In such situations, it is necessary to adapt the standard anti-COVID-19 modalities to make a visit safe. Other cases involve patients who are entitled or are without the possibility of being transported to an eye clinic. These cases are handled as needed and by clinical case. It is necessary to obtain information about the patient before making the visit; often this happens through intra-hospital computer networks where the examinations carried out are uploaded. Patients in a state of unconsciousness or semi-consciousness with assisted ventilation often encounter corneal problems such as epithelial defect, dry eye disease or corneal ulcers. Facial trauma is a common presentation to emergency departments, either alone or in combination with other injuries. Of overall facial fracture presentations, 36.3% of these includes trauma to the orbit. Ophthalmologist and surgeon can have a meeting prior to the control of the patient to analyze the management of the clinical case through video call or messages, in order to limit contact and save time by achieving the same result. Even more useful to having an adequate eye examination is uploading images to allow the ophthalmologist to set up a diagnostic hypothesis. For this task, telemedicine is widely used in this area [7,39,40].

#### 3.2.12. Distance Learning

Ophthalmology education via video conferencing or online courses can provide skills and training without conventional travel expenses. Telesurgery can also be used in ophthalmology with the possibility of operating the patient by videoconference with the aim of instructing/training the surgeon at the other end of the terminal. Store-and-forward telemedicine has a data storage and a sharing system that work asynchronously, landing medical information to a site, storing it digitally, then transmitting it to another place where it can be stored again before analysis. E-mail is a type of storage and shipping system. Real-time systems work synchronously. Speakers, microphones and cameras, audio and video conferences and group rooms also pass information at almost simultaneous speeds. Hybrid systems mix real-time and store-and-forward telemedicine capabilities [7].

### 3.3. Patient Satisfaction and Acceptability

Various surveys have reported that patients find teleophthalmology a highly permissible method for DR screening and 82% of respondents preferred telemedicine compared to in-person visits. Boucher and collaborators reported that patients consider tele-ophthalmology a highly admissible method for DR screening. Patients were asked to answer a questionnaire related to satisfaction after an in-person visit or after teleconsultation. In total, 99% of them were satisfied with telemedicine and 82% of them preferred it to an in-person visit. The explanations included absence of pupil dilation, short amount of time required for the examination, less annoyance and ease of healthcare accessibility. Only 3.2% did not trust telemedicine. In another study, a video slit-lamp, an automated perimeter, a non-mydriatic fundus camera and a videoconferencing system were installed in a healthcare site located in a rural area. In total, 29 patients were examined on the rural site, whereas 41 control patients were examined in the urban hospital. Patients in both groups were equally satisfied, but 96% of patients who visited the rural center wanted to continue to be followed-up at the same site due to reduced travelling, expenses and time. Among health professionals, younger doctors are more likely to use and familiarize themselves with technology, which could be a problem for older patients [3,4,7,19]

### 3.4. Benefits of Telemedicine

The costs of telemedicine have been shown to be lower than the physical management of the patient. In fact, the ease of access to treatment and the speed of patient evaluation saves on transport costs to go to the health facilities, avoids long waiting lists and to reduces disposable hospitalizations. Following the COVID-19 pandemic, telemedicine played a central role in patient management. It has been possible to limit infections by offering telephone consultations such as COVID-19 triage in order to better frame the case of the patient in question. Even general medicine, similar to the studies of specialists, has used a management method based more on telemedicine to better organize patient visits and avoid gatherings in waiting rooms. The foundations have been laid for a telematic organization which could become routine in the future [5,14,39].

### 3.5. Disadvantages of Telemedicine

Telemedicine potentially introduces a new form of medical error: a risk that the doctor could become an automated figure linked to remote management. In addition, physical contact with the patient can be lost and there is a risk that it will become just a simple case to be managed online and the patient himself realizes that he is evaluated less carefully than he would be during an in-person visit [1,41]. The difficulties of application are the costs of telemedicine services for the purchase of devices, the difficult technical management between different countries and the lack of defined ethical rules and regulations. Different works show that legislation on telemedicine activity is lacking if not completely absent in many cases [38,42].

### 3.6. Medical Prescription

In the process of medical examination, in general, the final step is the prescription of a medical therapy to treat or manage the patient’s condition. When the approach is electronic, problems may arise in the prescriptions. Currently, there is varied regulation regarding online prescriptions [3]. The guidelines published by the Medical Council of Hong Kong (MCHK) explicitly advises doctors to prescribe medicines electronically only to patients who have previously visited in-person, as they have adequate knowledge of the patient’s clinical condition, and only after ascertaining that the drug serves the patient’s needs [4]. In one of the first cases of telemedicine in California, Dr. Hageseth, a psychiatrist in Colorado, was criminally charged with prescribing unlicensed medicine in California. He had prescribed medication on the Internet to a patient who then committed suicide. Following the trial, he was sentenced to 9 months in prison. However, it should be pointed out that electronic prescription in telemedicine avoids errors, such as in the tax code, the age of the patient or the dosage, and also because of the incomprehensible calligraphy of the doctor [5].

### 3.7. Legal Aspects

As for other technology-driven innovations, the legal and regulatory framework has evolved more slowly than this new model of care delivery [4]. It is important to point out that this new way of interaction between the parties can bring together environments over which there is a different jurisdiction. For example, in the United States, in 2016 the American Medical Association provided new guidelines for ethical practices in telemedicine. Physicians practicing telemedicine must ensure that appropriate protocols were followed to care the safety and integrity of patient information [3]. Thirty-six states and the District of Columbia have equality laws covering private insurers and reimbursement for tele-sanity services. The federal government places numerous limitations on medical reimbursement for tele-sanity services, based on the geolocation of the patient and doctor, as well as the type of facility.

To increase and implement the propagation of telemedicine in common use globally, it is essential to standardize telemedicine laws. An important inducement, favored by WHO, is found in *Recommendations on Digital Interventions for Health System Strengthening,* which states <<Put–telemedicine-systems in place to ensure data privacy, ownership, access, integrity and protection of patient information. Ensure that these systems meet national legal standards [3,43]. Security is needed to address not only risks to patient confidentiality, but also risks to data integrity such as unauthorized alteration of data>> [43]. The document itself represents both an incentive for the development of telemedicine as well as a limiting and less universalizing key guide, reserving wide discretionary power to individual states.

In the same American landscape, a physician cannot obtain a universal professional licence valida to use all over the country; he can only operate in the same state that releases the license itself. This aspect led an important limit to interstate telemedicine practices. To facilitate the circulation and the usability of tele-health service, in 2014 the Federation of State Medical Board approved the Interstate Medical Licensure Compact, which allows for an expedited licensure process [44].

For maximum data protection, many authors point out the necessary uniform guidelines, similar to the GDPR enforced in the European Union [3]. The EU has increased the presence of telemedicine in Europe as a standard medical service for European patients protected by the social security system. Regulation 2016/679 represents European data protection legislation. With the Treaty of Lisbon, the protection of personal data is a fundamental right in the EU. With the European regulation, we move from a proprietary vision of the data (not negotiable without consent) to a vision of data control that improves free movement. The European Union of Medical Specialists has published a document that contains the “European Definition of Medical Law”. This is the approved version: “*The medical law covers all professional activities, for example of scientific technology, teaching, training, educational, organizational, clinical and medical, carried out in order to promote health, prevent diseases, make diagnoses and prescribe therapeutic treatments or rehabilitation treatments for patients, individuals, groups or communities, within the framework of ethical and deontological rules. The medical act is the responsibility of the qualified doctor; it must be performed by the doctor or under his direct supervision and/or prescription*” [37]. There is the fundamental right of patients to the confidentiality of their medical records. The Network and Information Security Directive for the protection of health data is the first step of the European cyber-security strategy. Approved by the European Parliament on 6 July 2016, the directive aims to strengthen cyber security and resilience within the Old Continent. Telemedicine creates problems resulting from the involvement of non-clinical staff in tele-consultations and the vulnerability of transmission lines to security breaches. Health data on patients’ intimate details are subject to a general ban on dissemination, as well as increased protection. According to the European regulation, health data can only be used for health-related purposes (therapeutic purposes), for the supervision of the National Health System and for research in the public interest. In this sense, in Italy, with the new Privacy Code, additional measures have been provided to protect health data, established by the national control authority and revised every two years. The Italian legislator, with the decree updating the Privacy Code, has introduced the possibility that the Guarantor imposes specific guarantee measures (additional to normal security measures) for the processing of health data. The Italian Ophthalmological Society has developed informed consent forms. Before proceeding with the collection of data, it is necessary to inform the patient. The document on informed consent indicates the object of data collection, the purposes of the processing, the processing methods, the subjects to whom the data may be communicated, the identification data of the data controller and the methods of protection of his data. Therefore, it presupposes two distinct moments: information and actual consent. Article 1 of Law 219/2017, according to the principles set out in Articles 2, 13 and 32 of the Constitution, establishes that no medical treatment begins or continues without the informed consent of the interested party, except in cases expressly provided for by law, such as treatment for the public interest. The Italian legal system establishes the freedom to choose the place of treatment (Article 32 of the Constitution and Legislative Decree 502/92) and the freedom to undergo or not to undergo treatment (Article 1 of Law 219/2017). Several works show that legislation on telemedicine activity is lacking if not completely absent in many cases [24]. The Italian Ministry of Health underlined the importance of the use of telemedicine at national level to frame the guidelines, and during the COVID-19 pandemic, more than ever, a very fragmented picture appeared [45]. In the absence of direct physical interaction, it can be difficult for doctors, through telecommunication tools, to verify the identity of patients (especially new patients) and establish a solid doctor –patient relationship based on trust and respect. Ophthalmologists who work as remote doctors should receive initial training and continuous updates to ensure good clinical practice; in fact, the virtual visits, although they have been a valuable public health tool in the COVID-19 emergency [30], have shown how essential it is to have training for health professionals aimed at mastering the chosen technological platform, the optimization of lighting and the positioning of the cameras [46].

Given the above, it could be more prudent for surgeons to select only patients with whom they are familiar, such as patients who have undergone surgery, patients who need chronic management over time and with whom they have already come into contact etc. As a general rule, surgeons are advised to use their clinical acumen to decide whether and how to use a remote assessment, whether as a single visit or only at completion, whether it is appropriate in a particular situation, and whether telemedicine will allow surgeons to meet the necessary legal standard of care [4]. The progressive shift in health care, from present to virtual, leads to a greater risk of misdiagnosis [1]. For this reason, ophthalmologists must provide insurance coverage for liability for medical negligence for this health activity [38].

From the regulatory point of view, in Italy, the first provision dedicated to the implementation of Telemedicine services is represented by the Agreement between the Government and the Regions on the document containing “*Telemedicine-National Guidelines*”, approved by the General Assembly of the Superior Council of Health and by the Conference of State Regions. The guidelines are intended to respond to the need to standardize the experimental initiatives activated in the area and to outline a global framework that also identifies the priorities for the use of Telemedicine. The document recognizes the substantial equivalence between the services provided in telemedicine and traditional services. Due to several advantages related to digital health, the government has decided to adopt a strategy coordinated and shared at the national level for the introduction of digital health. Three documents are currently the main references for the development of Telemedicine services in the area: the Health Pact for the years 2014 to 2016; the Pact for Digital Health and the National Chronicity Plan. With regard to the reimbursement of services and benefits, Legislative Decree 502/12 presupposes that “the types of assistance, services and health services that present, for specific clinical conditions or risks, scientific evidence of a significant benefit in terms of health, at an individual or collective level, in relation to the resources used”. Telemedicine services can be classified into 3 macro categories: (i) Specialized Telemedicine; (ii) Tele-health-care and (iii) Tele-assistance. Telemedicine can promote a reduction in hospitalization of chronically ill people, admissions to nursing homes and the mobility of patients seeking better care. The evolution of telemedicine poses a number of legal problems ranging from authorization and accreditation profiles to those relating to the protection of patient privacy. The use of information technologies in the health sector is not directly evaluated by Law 24/2017, which deals with the provisions on the safety of care and the assisted person and the professional responsibility of health workers. In terms of professional responsibility, it is necessary to establish that the telemedicine service is comparable to any health service for diagnosis or treatment. Three categories of subjects can be noted: the service center; the health facility providing the telemedicine service and the healthcare professional [37] (Figure 2).

#### 3.7.1. Informed Consent

On the subject of informed consent in Italy, the reference legislation is provided by the Law, n. 219/2017 [45], although the MCHK guidelines do not foresee specific written consent. For the visit in telemedicine, it was established that all necessary information related to telemedicine must be fully explained to the patient in a clear and understandable way, in order to obtain free and informed consent. In the informed consent document, there is information about the subject who acquires data, how these data have been treated, the aims of the treatment, communications to third parties and data protection [38]. In accordance with the aforementioned legislation, informed consent must be written or provided through audio-video recordings and must be inserted in each medical record. Obtaining informed consent is also a prerequisite for treatment provided by telemedicine [45]. In this regard, it should be mentioned that the Italian Society of Ophthalmology developed a customized informed consent statement. Informed consent is a means to ensure that the patient is aware of all aspects of the proposed treatment (indications, expected side effects, reliable results, etc.) [38]. Additionally, the patient should be informed about the risk of breaches of their personal data confidentiality and technological malfunction, and also about other existing treatment options during the practice of telemedicine, with the aim of lowering the risk of lawsuits against health professionals and health facilities [1]. In digital healthcare delivery, the importance of communication and informed consensus can be undermined, considering that telemedicine software is often non-transparent because they do not know how their data will be processed [2]. The Federation of State *Medical Boards* stated that *“the relationship (doctor-patient) is clearly established when the former agrees to undertake the visit and set up a therapy and the patient agrees to be treated”*, the patient’s consent runs counter to medical paternalism. Doctor-patient communication must be adequate to the patient’s condition and capacity while being current and complete. The communication time between doctor and patient constitutes treatment time aimed at building a preventive therapeutic alliance. One limitation in the use of teleconsultation is the impossibility of performing an accurate physical examination. While superficial observation of the skin/eye or any wounds in a postoperative patient or other procedural examinations could be carried out by digital devices, the evaluation of pathologies of surgical interest such as the acute abdomen or a simple physical examination would be very difficult in current practice. It is therefore of crucial importance that both patients and doctors are conscious of these methodological restrictions; in these cases, an in-person visit is clearly necessary. In any case, the completeness of activities carried out through telemedicine should be equal to traditional approaches [4].

#### 3.7.2. Privacy Issues

The ability to track all the checks and visits carried out, the medical prescription, documents and to archive the acquisitions of data in order to evaluate over time the evolution of a clinical picture is a great strength of telemedicine. To date, this information can allow the creation of extremely sensitive data files from a legal point of view. Selfies and photographs taken with mobile phones, nowadays, are sent to healthcare professionals for medical advice. These can be sent at the initiative of the patient to doctors, on the instruction of the attending physician or a practice (widespread in recent times) or also to social media groups [4]. These photographs and accompanying information all fall under the definition of “health data”, which is all sensitive information concerning health, physical and mental conditions of a person (including genetic data) [38] that should be inserted in the patient’s medical record and should be safely transferred and stored securely to ensure patient data confidentiality [6]. There are few references in the literature on how to face legal and ethical issues; in particular the responsibilities of the patient and the doctor on the sharing of data. There are currently no guidelines or protocols on how to keep records, nor on how to handle images and other types of data [4]. It is precisely for this reason that data privacy becomes a key point in the broad framework of telemedicine. In accordance with the European legislation on the protection of personal data within the EU Reg. 679/2016, those who provide the platforms for sharing sensitive data should ensure their security. At the same time the doctor should pay attention to the sharing of clinical data for the purpose of counseling. According to surveys on various samples, many agree on the need to regulate the privacy of the sharing of this data. Tele-health operators must create appropriate communication links with other intermediary healthcare providers in the production of services in order to avoid conflicts over patient management and minimize communication errors [1]. In accordance with the aforementioned Law, it is important to appoint the data protection operator responsible who also ensures the compliance of the processing activities with the law. The operators involved are also required to handle sensitive data correctly, both when storing them and when transmitting them to third parties, especially when using online storage systems. This may also require a management model based, for example, on the use of data encryption tools. [38]. Lastly, it is essential that the activities carried out comply both with the regulations in force where the service is provided and with those where the patient is situated [4].

### 3.8. Future Perspectives

It would be appropriate to set a uniform standard and a clear process for obtaining medical licenses for doctors who practice online consultations. Furthermore, it would be necessary to harmonize the different jurisdictions. Medical coverage of tele-sanity services, including remote monitoring of patients, should be extended beyond rural areas. The next generation of healthcare professionals need to be well educated on how to integrate telemedicine into their clinical practices [4]. Specifically, in ophthalmology, it would be useful to create a common system in which the ophthalmologist is able to quickly get in touch with the patient before seeing him in person, in order to be able to frame the severity of the problem and direct the patient in a rapid management of the case. In this way, you would save action time and maximize the result. Moreover, it would be useful to create a platform, or implement the current ones already present, in which ophthalmologists can quickly exchange consultations with each other in order to better frame the clinical case. A limit would be the international relations between doctors, and for this reason, it is necessary to unify the legislation in force between the various countries to minimize the differences that are currently present.

Many features of telemedicine could cause problems, namely violations of privacy, physical distance, inclusion of new technological advances and weakening of the doctor/patient relationship. Therefore, healthcare professionals should consider that telemedicine could cause further potential medical failures within a standard of care that, in some cases, can be more challenging [46].

## 4. Conclusions

Undoubtedly, the COVID-19 epidemic has changed the medical practice and accelerated the introduction and development of telemedicine [47].

Ophthalmology represents the field where telemedicine has spread, according with technological approaches characterizing this specialty. Visual examination is the key element which joins advanced diagnostics and treatment technology systems, and which may also be pursued through a smartphone. The ophthalmological world offers a large opportunity to telemedicine: diseases of orbit, appendages and ocular surface are easily diagnosable through visual signals, as digital images of the pupils’ light reflections can help ophthalmologists assess astigmatism and strabismus. The pandemic era allowed telemedicine to also follow patient outcomes during intravitreal injections into the management of macular oedema. In post-surgical assessment, through telemedicine we can evaluate a patient after cataract surgery. In rural areas, tele-ophthalmology helps to assess loss of vision from one eye, acute pain, traumas, speeding up the treatment or the patient sending to the nearest emergency room center. A further application in emergency setting is represented by the evaluations of a patient in a state of unconsciousness or semi-consciousness, which often creates corneal problems or facial traumas. The prevention role of tele-ophthalmology is another valid challenge for the screening of diabetic retinopathy and glaucoma. The perioperative management of surgical patients is expected to gain further popularity beyond the COVID-19 scenario.

As telemedicine is growing rapidly, operators need to acquire new skills in terms of communication and technology in order to provide remote care successfully. Although it offers the benefits of saving patient’s time and costs, it is necessary to make known limitations, including the absence of physical examinations, the possibility of transmission failure and risks of violations of privacy and confidentiality [3].

The medico-legal scenario shows remarkable complexity. On a hand, it can be attributed to the lack of strict regulations and directives from international associations; on the other, which also justifies the first consideration, the legal system of various states presents an organizational complexity. As a consequence, sometimes the individual territorial entities enjoy regulatory-autonomy which makes a superstructural project unworkable, or susceptible to minor modifications otherwise. The result is a complete fragmentation of the legal scaffolding which, often and willingly, stimulates the development in a different way without a unifying perspective, considering telemedicine in the various American states. It is useful to use the standards and guidelines set for clinical practice, training and research, and to use those developed by professional organizations such as the American Telemedicine Association (ATA), as well as those developed by their own institutions [48].

In summary, due to the disparities between European legal systems and beyond and the consequent difficulties in developing a common framework in many respects, the European Union can aspire to create a legal framework for the areas where European law pre-exists and adapt it to the specificities of telemedicine [49].

## Figures and Tables

**Figure 1 ijerph-19-05614-f001:**
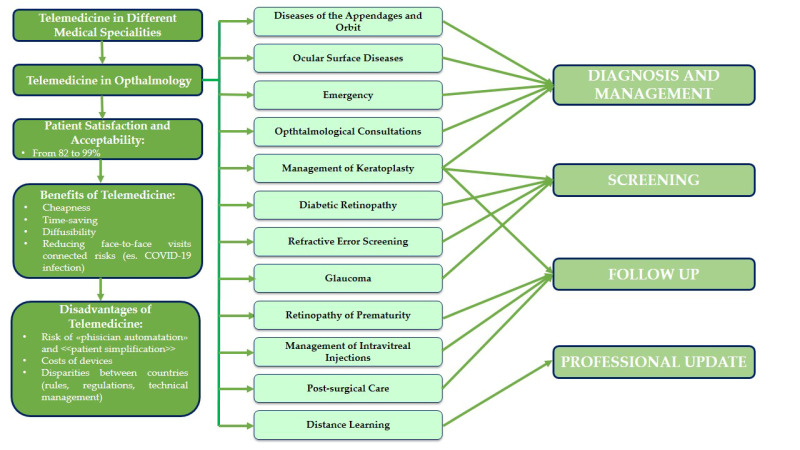
Flow chart. The figure analyzes the first part of review concerning all ophthalmological applications of telemedicine.

**Figure 2 ijerph-19-05614-f002:**
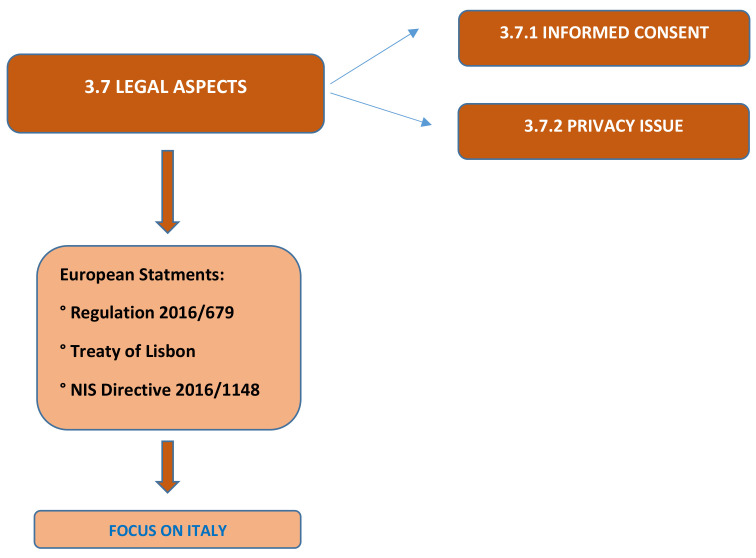
Flow chart. The figure analyzes the second part of review concerning all medico-legal implications of telemedicine. Numbers in each box correspond to the corresponding paragraph.

**Table 1 ijerph-19-05614-t001:** Evaluation of the main ophthalmological clinical studies treated in the manuscript.

Author (Year) [Ref.]	Aim	Sample Size	Outcomes
Meshkin RS [21]	To evaluate the efficacy of telemedicine for emergency triage	129 patients	Telemedical examination may be useful to reliably triage eye disease
Silvia PS (2015) [22]	To evaluate the ability of trained nonphysician retinal images to perform DR evaluation at the time of UWF imaging in a teleophthalmology program	3978 eyes (1989 patients)	The telemedicine programs for diabetic retinopathy increase the rate of retinal assestments
Daniel E (2015) [23]	To describe a centralized system for grading ROP digital images	5520 images sets	The e-ROP training system is reliable to detect potentially serious ROP
Maloca P (2018) [24]	To study a novel home.monitoring OCT device in aged-macular macular degeneration	31 patients	This device may facilitate the monitoring of chronic retinal diseases
Tsapakis S (2018) [25]	To present a home-based visual field examination method compared to the Humphrey perimeter	20 eyes	The home-based visual field test exhibits a reasonable receiver operating characteristic curve when compared to the Humphrey perimeter
Alabi RO (2019) [26]	To evaluate telemedicine in the evaluation of the recovered donor corneas	11 samples	The evaluation of digital images is adequate for identification of specific corneal findings

ROP, Retinopathy of premature; DR, Diabetic Retinopathy; UWF, Ultra Wide-Field; OCT, Optical Coherence Tomography.

**Table 2 ijerph-19-05614-t002:** The time interval proposed for ophthalmological examinations [33].

	General Risk	Ophthalmological Risk	Interval
Low general risk	Low	No	2 years
Other risk constellations	High	No	1 year
Unknown general risk	Unknown	No	1 year
Diabetic retinal changes		Yes	1 year or less
Diagnosis of type 2 diabetes after the age of 10, or 5 years after the onset of type 1 diabetes			Short-term
Visual worsening, distorted vision, blurry vision, spots before the eyes			Short-term

## Data Availability

Not applicable.

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
