# Peer review of "Applications and Current Medico-Legal Challenges of Telemedicine in Ophthalmology"

_ijerph, 2022, doi:10.3390/ijerph19095614_

Round 1
Reviewer 1 Report
Thank you for giving me an opportunity to review this paper. It is a well-written paper; however, there are several issues that need to be addressed.
- Why did authors only search one database? I suspect, the authors might miss a lot of potential studies. They should also search Scopus, Embase and Web of Science databases.
- The method part is too short. Please describe more about your study inclusion, exclusion criteria, and information extraction process.
- Please cite, 22 original articles and 4 reviews, and a careful narrative view here.
- Please provide privacy and legal issues in chronological order like 1,2,3... so it will be easy to absorb information from this paper.
Reviewer 2 Report
To the part of ophthalmology:
Retinopathy of prematurity is one of the leading causes of childhood blindness. Which methods can be available for telemedical screening? What are the advantages/ disadvantages of it?
Diabetic retinopathy: Which scales were used in the articles to determine whether the patient should be immediately/ within a week/ within a month etc referred to an ophthalmologist; or whether the case is urgent or not urgent: Summarize this problem shortly in a table.
In some countries, optical coherence tomography is available for patients (esp. with age-related macular degeneration) at home. What could be the importance of it?
Measurement of eye pressure is not enough for the detection of glaucoma progression, optical coherence tomography and visual field examination are required: which recommendations can be found in the articles for prevention of progression?
Reviewer 3 Report
This is an article entitled “Applications and Current Medico-Legal Challenges of Telemedicine in Ophtalmology (ijerph-1688424)” which reviews the role and the main applications of telemedicine in the ophthalmic clinical practice and related medicolegal aspects.
The English needs revision.
Abstract
- Good.
Introduction
- Good.
Methods
- Good.
Results
- Please also include and discuss the influence of telemedicine in keratoplasty patients as well.
- Please also discuss the available web sites for self visual acuity check.
Conclusions
- Good.
References
- Okay.
Reviewer 4 Report
The digital revolution is redesigning the healthcare model, and the telemedicine offers a good example of the best cost-effectiveness ratio. The COVID-19 pandemic has catalysed the use of the telemedicine.
The authors proposed a review aimed to describe and discuss the role and the main applications of telemedicine in the ophthalmic clinical practice as well as the related medico-legal aspects.
They considered 22 articles and 4 reviews focused on this topic and published in English language from 1998 and 2021 from the online database of Pubmed by using the keywords: ‘telemedicine’, ‘privacy’, Ophthalmology, ‘COVID-19 and ‘Informed consent’.
The study generally highlighted that (a) Telemedicine is able to guarantee patient care using information and communication technologies. (b) Technology gives the opportunity to link doctors with the aim of assessing clinical cases, maintaining high standards of care while performing and saving time as well. (c)Ophthalmology is one of the fields in which telemedicine is most commonly used for patient’s management.
The authors concluded that Telemedicine offers benefits to patients in terms of saving time and costs and avoiding physical contact; however, it is necessary to point out significant limitations such as the absence of physical examinations, the possibility of transmission failure and potential violations of privacy and confidentiality.
The review has merits. However it needs some improvements
Please reply to the following comments
- It seems that there are two key questions that the narrative review must answer, “The goal of this review is to highlight the potential role of telemedicine as a future resource in the diagnosis and management of various ocular disorders, and to further compare the legal regulations in force in different countries.” Please explicit them better.
- Better details in the methods how you answered to the two key questions?
- Have you used specific parameters with a scoring system to assess each manuscript?
- Have you used a checklist to assure a quality during the phases of the review?
- You cite some keys. It is important to understand how you combined the keys to trace the construction of the study.
- You have divided the narrative review into some themes. Please justify this and put a flow chart to aid the reader.
- Some tables with the references could help the reader
Round 2
Reviewer 1 Report
Thanks for the revised version.
Reviewer 4 Report
The authors improved a lot the manuscript.
There are not futher comments.